# The Effects of *Morus alba* L. Fortification on the Quality, Functional Properties and Sensory Attributes of Bread Stored under Refrigerated Conditions

Joanna Kobus-Cisowska [1], Marcin Dziedziński [1,*], Daria Szymanowska [2,3], Oskar Szczepaniak [1], Szymon Byczkiewicz [1], Aleksandra Telichowska [1] and Piotr Szulc [4]

[1] Department of Gastronomy Sciences and Functional Foods, Poznan University of Life Sciences, 60-637 Poznan, Poland; joanna.kobus@up.poznan.pl (J.K.-C.); oskar.szczepaniak@up.poznan.pl (O.S.); szymon.byczkiewicz@gmail.com (S.B.); a.telichowska@gmail.com (A.T.)

[2] Department of Pharmacognosy, Poznan University of Medical Sciences, 60-781 Poznan, Poland; daria.szymanowska@up.poznan.pl

[3] Department of Biotechnology and Food Microbiology, Poznan University of Life Sciences, 60-627 Poznan, Poland

[4] Department of Agronomy, Poznan University of Life Sciences, 60-632 Poznan, Poland; piotr.szulc@up.poznan.pl

* Correspondence: marcin.dziedzinski@up.poznan.pl; Tel.: +48-61-8486330

**Abstract:** Mulberry is one of the most beneficial plant of our planet for sustainable development. White mulberry (*Morus alba* L.) is widely recognized for its health-promoting properties. It is characterized by a high content of bioactive compounds, mainly flavonoids, and has a strong antioxidant effect, and thus can have a beneficial effect on health. The aim of the study was to evaluate the effect of freezing storage of bread with the addition of extract from mulberry leaves and fruits on the content of polyphenols, antioxidant activity and sensory properties. The stored bread with mulberry addition was characterized by high content of phenolic compounds, reducing and chelating activity and antiradical activity. The addition of mulberry had greater effect on the increase in the content of protocatechuic and chlorogenic acids, and isoquercetin among the flavonols. Bread enriched with mulberry was microbiologically clean and sensory accepted both after baking and after 30 days of storage under refrigerated conditions. White mulberry is a raw material which can be used as an addition to enrich refrigerated bread. The use of the extract and mulberry fruit to fortify bread is consistent with the principle of sustainable development due to the use of raw materials which are a good source of compounds contributing to the improvement of the well-being of the population.

**Keywords:** white mulberry; *Morus alba*; antioxidants; polyphenols; bread

## 1. Introduction

Mulberry is regarded as multipurpose plant due to recognition of its role in environmental safety approach, as a medicinal plant and its industrial uses in various sectors [1]. The broad ecological adaptability of mulberry to different environmental conditions such as temperature, water, and soil, and enables it to have multiple ecological protective functions in water and soil conservation, wind resistance and sand consolidation, water source preservation and air refreshment [2]. The health-promoting effect of white mulberry (*Morus alba* L.) results from the presence of many compounds with biological activity. The literature indicates the presence of polyphenolic compounds, such as, quercetin, kaempferol, rutin, isoquercetin, astragalin, myricetin derivatives and other glucosides [3]. Mulberry preparations are also a source of tannins and coumarin compounds (scopolin

and skimin). The presence of numerous phenolic acids (chlorogenic acid, caffeic acid, hydroxybenzoic or ferulic acid) was determined in mulberry leaves and fruits [4]. Furthermore, mulberry was shown to contain terpenes such as citral, linalol acetate, linalol or cis-3-hexen-1-ol, and steroids such as β-sitosterol [5–10]. From the pharmacological point of view, the important active ingredients of mulberry are alkaloids which may affect carbohydrate metabolism: 1,5-dideoxy-1,5-imino-D-sorbitol (DNJ) and its derivatives. The variety and specificity of the compounds contained in the leaves, roots, fruits and bark of mulberry has an effect on the antioxidant and health-promoting activity in model and biological systems, which has been repeatedly demonstrated in the available literature [11].

Cereal products, including bread, provide components necessary for the proper functioning of the body such as proteins, saccharides, minerals and biologically active substances (e.g., dietary fiber and vitamins). Fresh bread is a short-life product, so freezing the dough is a common practice in industrial production due to its many economic advantages. Freezing of bread saves time and labor during production and extends the shelf-life of the product, which facilitates product distribution [12]. However, the process of freezing and storage affects the structure and quality of the dough, mainly due to the formation of ice crystals, recrystallization and redistribution of water in the product [13]. These processes negatively affect the quality of the dough and lead to an increase and destruction of the crumb pores, weakening of the gluten network and deformation of the starch grains, thus there is a search for ways to reduce the adverse changes associated with the cooling processes [14].

Bread fortification is a common practice used to prevent malnutrition and nutritional deficiencies in populations and adding functional properties, but additional ingredients can also influence other parameters, such as appearance, flavor and taste [15,16] Presence of various ingredients in bread, such as plants' components and extracts, can also influence the physicochemical stability and quality parameters during storage [17,18]. In studies in which bread was enriched with *Garcinia mangostana* fruit extract, it was shown that such an addition increases the attractiveness of the bread in terms of its functionality, while reducing the bread volume [19]. Sun-Waterhouse et al., by adding extracts of dried apples, kiwi and blackcurrants to bread, showed that they can be a beneficial functional addition [20]. Peng et al. proved that the antioxidant properties of bread can be improved by adding grape seed extract rich in catechins and epicatechins [21]. In another publication, it was shown that *Cistus* extract contributes to the improvement of the microbiological quality of bread [22]. The addition of *Cistus* extract influenced the content of polyphenols by increasing it from 8.88 (wheat bread) to 78.71 mg/100 g (bread with 7.5% cistus extract). The literature also indicates other raw materials that can be used to enrich bread, such as hops, rice, blueberries, etc. However, there is no data in the literature on the impact of refrigerated storage on baking or ready-made bread on the functional properties resulting from the addition of white mulberry. Using mulberry leaves to enrich bread is consistent with the principle of sustainable development due to the use of raw materials that are a good source of compounds contributing to the improvement of the well-being of the population. Mulberry leaves are currently not a component of many foods for humans and are available in many locations around the world. Bread enriched with ingredients derived from white mulberry and the good quality of such bread after storage in a frozen state may indicate new directions in reducing food waste by extending the shelf life of bread that maintains high sensory and microbial quality and functional properties. The aim of the study was to investigate the effect of additives in the form of mulberry leaves and fruit extract on the quality and properties of fresh and frozen bread. Enriched and control bread products were compared in terms of sensory characteristics and polyphenol content, as well as antioxidant and microbiological activity.

## 2. Materials

The material for the study were the leaves of white mulberry *Morus alba* L, Wielkolistna Żółwinska cultivar, which were collected from the plantation of the Institute of Natural Fibers and Herbal Plants in Poznań, at the Experimental Plant Pętkowo (52°12′N 17°15′E). The leaves and peduncles were dried at 60 ± 1°C and ground to the form of powder with a grinding degree of 0.8–0.08 mm [3]. The dried

leaves were stored in containers (PE) without access to light at 4.0 ± 0.5°C. Mulberry fruits were imported from China (Zheijang, 30°15′N and 120°9′ E). The air drying at 65±1°C was applied.

## 3. Methods

### 3.1. Extraction

Industrial extract from mulberry leaves was obtained by extraction of leaves by a counter-flow method at a temperature of 80–90°C, with water in counter-flow in a quantity 10 times higher than the raw material. Vacuum concentration of the extract was performed at 70–80°C in 7.5–10 h at 0.6–0.8 atm. The concentration was completed when the extract reached 25–30% of dry substance. Then spray drying of the concentrated extract was carried out using a Niro Atomizer SR16 type spray dryer equipped with a disc as a spraying element. The capacity of the dryer was about 16 kg of evaporated water per hour. The drying was carried out with hot air in counter-flow. Inlet air temperature at 175–185°Cand outlet air temperature at 85–95°C were controlled. As a result of drying, a fine powder of dry herbal mixture extracts was obtained. The obtained extract in the form of powder was stored at room temperature without access to light.

### 3.2. Bread Recipe and Baking Process

The bread was baked using the direct, single-phase method with ingredients listed in Table 1. The dough was obtained by mixing the ingredients in a laboratory mixer GM-2 (ZBPP, Bydgoszcz). Fermentation at 37°C at 80% relative humidity lasted 1 h (with punching after 30 min). The dough was fermented in the same way as wheat dough, without using leaven. Pieces of dough weighing 250 g were placed in the fermentation chamber until optimal growth (37°C, relative humidity 80%). The baking was carried out for 30 minutes at 220°C in a convection oven. After baking, breads were allowed to cool down to room temperature for 3 h. Subsequently, the breads were sliced (slices about 1.5 cm thick), and kept frozen (−20°C) until analysis. Breads for analysis were thawed in the air at room temperature until they reached the temperature of 20°C. Bread with mulberry (*Morus alba* fruit) was determined as PM1, analyzed on the day of baking, and PM30 after 30 days of freezing storage, and bread without additives as control bread PK1 analyzed on the day of baking and PK30 after the storage.

**Table 1.** Composition of bread ingredients (%).

| Ingredients | PM | PK |
|---|---|---|
| *Morus alba* leaves extract | 1.0 | 0.0 |
| *Morus alba* fruit | 3.0 | 0.0 |
| Chia seeds | 3.0 | 3.0 |
| Wheat flakes | 2.7 | 2.7 |
| Sunflowers seeds | 2.0 | 2.0 |
| Cornflakes | 4.0 | 4.0 |
| Millet | 1.0 | 1.0 |
| Broken soya beans | 2.0 | 2.0 |
| Soybean meal | 1.0 | 1.0 |
| Linseed | 1.0 | 1.0 |
| Wheat malt flour | 1.0 | 1.0 |
| Sesame seeds | 1.0 | 1.0 |
| Acetic acid | 0.1 | 0.1 |
| Lactic acid | 0.1 | 0.1 |
| Citric acid | 0.1 | 0.1 |
| Sugar | 1.0 | 1.0 |
| Emulsifier E 472—mono and di fatty acid glycerides | 2.0 | 2.0 |
| Rape lecithin | 2.0 | 2.0 |
| Stabilizer E 466—sodium carboxymethylcellulose | 2.0 | 2.0 |
| Ascorbic acid | 2.0 | 2.0 |
| Wheat flour | 38.0 | 38.0 |
| Rye flour | 26.0 | 30.0 |
| Yeast | 3.0 | 3.0 |
| Low-sodium salt | 1.0 | 1.0 |
| Water | 66.0 | 68.0 |

PM—bread with addition of mulberry, PK—control bread without mulberry.

### 3.3. Qualitative and Quantitative Analysis of Phenolic Acids and Flavonoids

The qualitative and quantitative analysis of phenolic acids, flavonoids and ascorbic acid was carried out using a high-performance liquid chromatograph Agilent Infinity (Agilent Technologies, USA) according to the methodology of Kobus et al. 2019 [23]. The NovaPack C18 column (5 mm, 150 × 3.9 mm) was used to separate phenolic acids. The separation using a UV detector was monitored at 250 nm for p-hydroxybenzoic, protocatechuic, gallic and vanillic acids and at 320 nm for caffeic, chlorogenic, p-coumaric, ferulic and sinapic acids. In the analysis of flavonoid content, the separation was monitored at a wavelength of 370 nm. Myricetin, quercetin, rutin, hyperoside and isoquercetin, as well as kaempferol and astragalin were determined in the examined extracts. Phenolic acids and flavonoids were identified and quantified by comparing their retention times and peak areas with those of their standards.

### 3.4. Antioxidant Activity in Model Systems with DPPH and ABTS

The determination of activity with respect to DPPH radicals was performed based on the method described earlier by Kobus et al. [3]. The principle of the method was based on spectrophotometric measurement (Metertek SP-830, Taiwan) of the color of the reaction mixture, in which, depending on the antioxidative capacity of the extract under investigation, the free radicals of DPPH (1,1-diphenyl-2-pyrylhydrazil) were scavenged. Measurement of absorbance was performed at a wavelength of 517 nm. The antiradical activity was expressed in mmol Trolox per 1 g of extract dry matter and in quercetin equivalents (mmol/1 g of extract dry matter). The results were also expressed as EC50 (mg extract/ml) and AE coefficient (AE = 1/EC50). The determination of activity against ABTS cationic radicals was performed based on the method described by Re et al. (1999). Antiradical activity was expressed in Trolox and quercetin equivalents (mmol/1 g of extract dry matter), EC50 (mg extract/ml) and AE coefficient (AE = 1/EC50) were determined.

### 3.5. Chelating and Reducing Properties

Ferrous ion-chelating effects and reducing power of samples were estimated according to the method of Kobus-Cisowska et al. by colorimetric measurement [24]. For this colorimetric assay, 1 mL of sample, 0.1 mL of 2 mM $FeCl_2$ and 0.2 mL of the ferrozine reagent were added to each tube. The mixture was vortexed for ~60 s and left for 20 min at room temperature. Absorbance values were recorded ($\lambda = 700$ nm) using the Meterek SP 830 apparatus (Taipei, Taiwan). Deionized water was used as the control, and ferrozine was used as a reference.

### 3.6. Bread Microbiological Quality Evaluation

Microbiological tests were performed to determine the total number of aerobic mesophilic and psychrophilic bacteria, mesophilic yeasts and molds, the presence of Enterobacteriaceae bacteria, *Listeria monocytogenes* and *Salmonella* spp., *Staphylococcus aureus*, the number of aerobic *Bacillus* spp. and sulfate reducing bacteria. Determination of the total number of mesophilic and psychrophilic microorganisms was performed in accordance with the guidelines of PN-EN ISO 4833:1:2013. Determination of the number of molds and yeasts was in accordance with the guidelines of PN-ISO-21527:2:2009. Detection and determination of Enterobacteriaceae was in accordance with the guidelines of PN-ISO 21528-2:2005.

Determination of the number of microorganisms was performed using the Koch flooding method. The number of microorganisms was expressed as an arithmetic mean in the form of colony forming units in relation to 1 g of product (cfu/g). Microbiological analysis included detection of the presence and number of Listeria monocytogenes according to the guidelines of PN-EN ISO 11290-1:1999/A1:2005 and the presence of *Salmonella* spp. according to the guidelines of PN-EN ISO 6579:2003/A1:2007. Additionally, the count of indicator microorganisms, anaerobic spore-forming bacteria in Wrzosek's

medium, coliforms in lactose and diamond green medium, Enterococci in sodium azide and bromocresol purple medium, and coagulase-positive staphylococci in Giollitti-Cantoni medium were determined.

### 3.7. Sensory Evaluation

Sensory evaluation of bread samples was carried out in a sensory laboratory meeting the requirements of the PN-ISO 8589:1998 standard. The method of quantitative descriptive analysis (i.e., sensory profiling) was applied to the detailed sensory characteristics of these samples, which was performed by a 20-person team specially trained for this purpose. A total of 16 individual qualitative characteristics of taste, smell and color selected in the preliminary tests were evaluated. The intensity of each qualitative note was determined by means of a 10 cm unstructured linear scale with appropriate border markings. The obtained results were replaced by numerical values.

### 3.8. Statistical Analysis

The statistical analysis was carried out with the STATISTICATMPL 13.1 software from StatSoft. Basic descriptive statistics for individual parameters were performed. The results presented in the paper are an arithmetic mean of at least two series performed in three repetitions.

The comparison of mean values of the examined traits was made with the use of variance analysis for factorial systems with different number of observations, and intergroup differences were evaluated with Tukey's test or Spjotvoll's test (extended Tukey's test for unequal number of samples). The fulfillment of the assumptions of this analysis was also checked. Pearson correlation coefficients were calculated to assess the strength of the relationship between the tested samples. The significance of the correlation coefficient was checked with Student's T-test. Statistical conclusions were made at the significance level of $\alpha = 0.05$. In order to examine the shape and direction of the relationship between the studied activities, appropriate non-linear regression equations (logarithmic and exponential) were determined. The principal components analysis (PCA) method was used.

## 4. Results

### 4.1. Polyphenols Content

It was found that the quantitative composition of phenolic acids in the types of bread studied differed significantly (Table 2). The higher sum of acids was determined in mulberry bread than in the control sample, at the level of 228.31 μg/g and 146.94 μg/g, respectively. The dominant acid in mulberry bread was protocatechuic acid ($p \leq 0.05$), the content of which was 95.28 μg/g, which constituted over 40% of all acids in this bread. The chlorogenic acid content was determined at a lower level of 59.35 μg/g. It was also shown that the content of these acids was higher in mulberry bread compared to the control sample by 50% and 160%, respectively.

Higher levels of gallic and caffeic acids were determined in mulberry bread, and no differences were found in the amount of p-hydroxybenzoic and vanillic acid in PM and PK. It was shown that the addition of extract from mulberry leaves and fruits increased the content of flavonols in bread, which were determined at 201.74 and 49.07 μg/g, respectively. The dominant flavonol in the PM sample was isoquercetin, which accounted for over 75% of all flavonols. The content of isoquercetin in PK was four times lower than in PM, which, however, still accounted for over 65% of all flavonols. The second flavonol in terms of quantity was rutin, with a percentage share in PM and PK bread products of 10 and 20%, respectively. No quercetin and kaempferol were found in the control bread. Table 2 shows the values Δ being the difference between the content of particular polyphenols in mulberry bread and native compounds present in the control bread. The addition of mulberry had greater effect on the increase in the content of protocatechuic and chlorogenic acids, and isoquercetin among the flavonols. It was demonstrated that storage of bread in the frozen state did not significantly affect the reduction of compounds content ($p \leq 0.05$).

**Table 2.** Phenolic acids and flavonols content in bread during freezing with *Morus alba* leaves extracts and mulberry fruits.

| Phenolic Acid (µg/g) | PM 1 | | | PK 1 | | | PM 30 | | | PK 30 | | | Δ PM1-PK1 | | | Δ PM30-PK30 | | |
|---|---|---|---|---|---|---|---|---|---|---|---|---|---|---|---|---|---|---|
| Protocateuchic acid | 95.28 [d] | ± | 6.76 | 62.12 [b] | ± | 2.58 | 81.22 [c] | ± | 3.23 | 49.77 [a] | ± | 3.55 | 33.16 | ± | 2.51 | 31.45 | ± | 3.53 |
| P-hydroxybenzoic acid | 6.97 [b] | ± | 0.12 | 6.82 [b] | ± | 1.25 | 5.63 [a] | ± | 1.22 | 7.93 [c] | ± | 0.83 | 3.15 | ± | 0.00 | 2.30 | ± | 0.00 |
| Vanillic acid | 19.25 [a] | ± | 2.05 | 19.15 [a] | ± | 1.77 | 23.17 [b] | ± | 1.41 | 21.21 [b] | ± | 1.26 | 0.10 | ± | 0.00 | 1.96 | ± | 0.01 |
| Caffeic acid | 17.86 [b] | ± | 0.32 | 0.89 [a] | ± | 1.18 | 31.42 [c] | ± | 3.74 | 1.55 [a] | ± | 0.44 | 16.97 | ± | 0.83 | 29.87 | ± | 1.64 |
| Gallic acid | 12.33 [b] | ± | 1.91 | 0.00 [a] | ± | 0.00 | 17.31 [c] | ± | 1.63 | 0.00 [a] | ± | 0.00 | 12.33 | ± | 1.09 | 17.31 | ± | 2.63 |
| Chlorogenic acid | 59.35 [d] | ± | 1.99 | 22.07 [b] | ± | 1.15 | 44.21 [c] | ± | 2.88 | 8.71 [a] | ± | 0.36 | 37.28 | ± | 2.41 | 35.50 | ± | 2.13 |
| P-coumaric acid | 10.85 [a] | ± | 1.02 | 22.42 [b] | ± | 3.06 | 10.93 [a] | ± | 1.83 | 20.55 [b] | ± | 2.72 | 11.22 | ± | 1.11 | 9.62 | ± | 0.99 |
| Ferulic acid | 6.43 [a] | ± | 0.50 | 13.50 [b] | ± | 0.74 | 4.65 [a] | ± | 1.07 | 7.72 [b] | ± | 0.24 | 7.07 | ± | 0.87 | 3.07 | ± | 0.02 |
| Rutin | 20.27 [c] | ± | 1.10 | 9.35 [a] | ± | 0.97 | 18.77 [b] | ± | 1.55 | 6.44 [a] | ± | 0.34 | 10.92 | ± | 0.52 | 12.33 | ± | 0.64 |
| Astragalin | 4.28 [c] | ± | 0.13 | 2.44 [b] | ± | 0.10 | 1.97 [a,b] | ± | 0.75 | 1.24 [a] | ± | 0.01 | 1.84 | ± | 0.51 | 0.73 | ± | 0.02 |
| Hiperozide | 0.00 | ± | 0.00 | 0.00 | ± | 0.00 | 0.00 | ± | 0.00 | 0.00 | ± | 0.00 | 0.00 | ± | 0.00 | 0.00 | ± | 0.00 |
| Quercetin | 13.94 [b] | ± | 1.69 | 0.00 [a] | ± | 0.00 | 14.43 [b] | ± | 0.29 | 1.84 [a] | ± | 0.01 | 13.94 | ± | 0.66 | 12.59 | ± | 0.12 |
| Izoquercetin | 154.25 [b] | ± | 2.68 | 32.69 [a] | ± | 1.66 | 155.77 [b] | ± | 6.74 | 33.37 [a] | ± | 2.28 | 121.56 | ± | 5.76 | 122.40 | ± | 2.87 |
| Myricetin | 7.71 [c] | ± | 1.44 | 4.57 [a] | ± | 0.33 | 6.97 [b,c] | ± | 0.93 | 4.24 [b] | ± | 1.05 | 3.14 | ± | 0.13 | 2.73 | ± | 0.01 |
| Kaempferol | 1.38 [b] | ± | 0.02 | 0.00 [a] | ± | 0.00 | 1.92 [b] | ± | 0.02 | 0.00 [a] | ± | 0.00 | 1.38 | ± | 0.11 | 1.92 | ± | 0.01 |
| Izorhramentin | 0.02 [a] | ± | 0.00 | 0.02 [a] | ± | 0.00 | 0.02 [a] | ± | 0.00 | 0.02 [a] | ± | 0.00 | 0.00 | ± | 1.36 | 0.00 | ± | 0.81 |

The mean values marked different small letters in line indicate the significance of differences, values with this same letter in the row are not significantly different from each other ($p \leq 0.05$); PM1—bread with addition of mulberry analyzed on the day of baking; PM30—bread with addition of mulberry analyzed after 30 days of freezing storage; PK1—control bread without mulberry analyzed on the day of baking; PK30—control bread without mulberry analyzed after 30 days of freezing storage.

### 4.2. Antioxidant Activity of Bread

It was shown (Figure 1) that the activity of mulberry bread samples both immediately after preparation and after freezing was statistically significantly different from that of the control sample ($p < 0.05$). It was observed that the PM sample activity increased more than in the control sample, as evidenced by high coefficients in regression equations, and at 120 µg/ml it reached 8%.

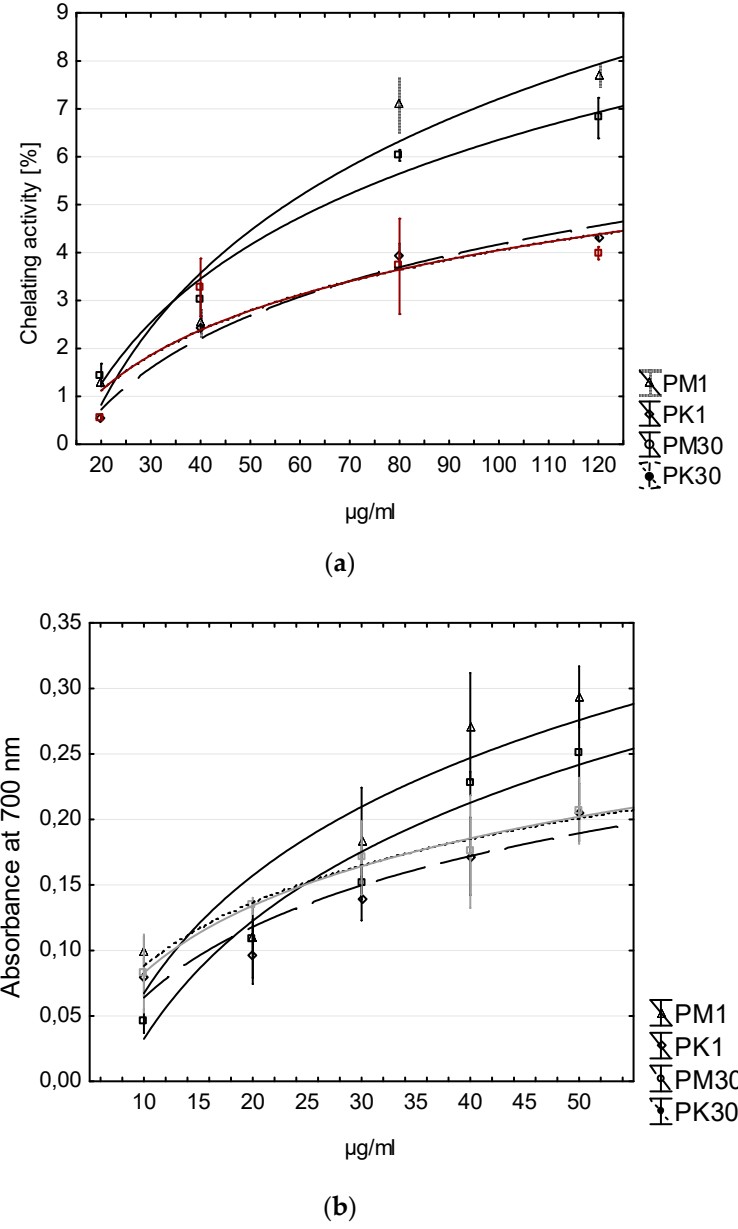

**(a)**

**(b)**

**Figure 1.** Chelating activity (**a**) and reducing power (**b**) of bread with extract of the leaves and mulberry fruit. PM1—bread with addition of mulberry analyzed on the day of baking; PM30 - bread with addition of mulberry analyzed after 30 days of freezing storage; PK1—control bread without mulberry analyzed on the day of baking; PK30 - control bread without mulberry analyzed after 30 days of freezing storage

The obtained regression coefficients confirm the results presented in the Figure: chelating activity initially increased linearly with the sample concentration and then stabilized at a constant level (Table 3). Breads with mulberry addition were characterized by a reducing power, which increased with increasing concentration. The value of coefficients *a* in the regression equations indicated

how much the value of the examined parameter (reducing power) changed when the value of x (concentration) changed by one unit. The results of the analyses indicate unequivocally that the addition of mulberry affected the reducing power of the examined bread samples.

**Table 3.** Coefficient values in regression equations presenting relationship between the concentration of the compounds from mulberry from bread with the addition of an extract of mulberry leaves and mulberry fruit and chelating activity with Fe (II) or reducing power.

| Sample | Chelating Activity $y = ax + b \times log10(x)$ | | | Reducing Power $y = ax + b \times log10(x)$ | | |
|---|---|---|---|---|---|---|
| | a | b | $R^2$ $p \leq 0.001$ | a | b | $R^2$ $p \leq 0.001$ |
| PM1 | −9.26 | 8.34 | 0.944 | −0.31 | 0.26 | 0.944 |
| PK1 | −5.55 | 4.61 | 0.935 | −0.26 | 0.13 | 0.954 |
| PM30 | −10.47 | 7.27 | 0.965 | −0.19 | 0.24 | 0.924 |
| PM30 | −4.11 | 3.11 | 0.934 | −0.12 | 0.19 | 0.975 |

It was demonstrated that the addition of mulberry preparations has a statistically significant effect on antiradical activity (Figure 2). Moreover, no differences in activity with respect to DPPH and ABTS were demonstrated between the samples after storage. This means that compounds responsible for antiradical activity present in bread were not inactivated during freezing.

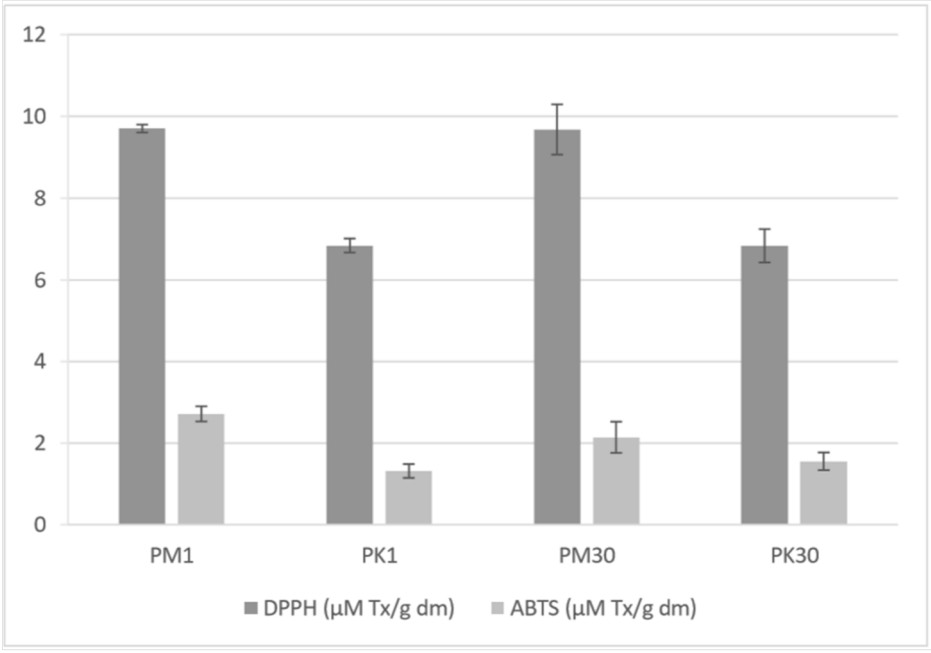

**Figure 2.** Antiradical activity measured by test with DPPH· and ABTS·+ radicals of bread with extract of the *Morus alba* leaves and mulberry fruit before and after storage. PM1—bread with addition of mulberry analyzed on the day of baking; PM30—bread with addition of mulberry analyzed after 30 days of freezing storage; PK1—control bread without mulberry analyzed on the day of baking; PK30—control bread without mulberry analyzed after 30 days of freezing storage.

The antioxidant effect of white mulberry extracts in the newly developed bread was the result of the presence of phenolic compounds (Table 4). The results of statistical analysis showed a positive correlation between the sum of flavonols and the sum of phenolic acids, and the level of DPPH radical inactivation (r = 0.83 and r = 0.89), ABTS (r = 0.97 and r = 0.76) and the reducing power (r = 0.81 and r = 0.86).



**Table 4.** Correlation coefficients between antioxidant activity indicators of bread and phenolic ingredients.

| Compound | Correlation | | | |
|---|---|---|---|---|
| | DPPH$^\bullet$ | ABTS$^{+\bullet}$ | Chelating Activity | Reducing Power |
| Rutin | 0.77 * | 0.94 * | 0.85 NS | 0.72 * |
| Astragalin | 0.85 * | 0.87 * | 0.38 NS | 0.82 * |
| Hiperozide | 0.53NS | 0.75 NS | 0.34 NS | 0.57 NS |
| Quercetin | 0.83 * | 0.95 * | 0.79 * | 0.79 * |
| Izoquercetin | 0.81 * | 0.98 * | 0.43 NS | 0.77 NS |
| Myricetin | 0.83 * | 0.81 NS | 0.42 NS | 0.62 NS |
| Kaempferol | 0.33 NS | 0.78 * | 0.44 NS | 0.77 NS |
| Izorhramentin | 0.84 * | 0.96 * | 0.45 NS | 0.79 * |
| Total flavonols | 0.83 * | 0.97 * | 0.42 NS | 0.81 * |
| Protocateuchic acid | 0.77 * | 0.89 * | 0.27 NS | 0.84 * |
| P-hydroxybenzoic acid | 0.04 NS | 0.09 NS | 0.11 NS | 0.38 NS |
| Vanilic acid | 0.12 | 0.22 NS | 0.68 NS | 0.22 NS |
| Caffeic acid | 0.87 * | 0.67 * | 0.44 NS | 0.69 * |
| Gallic acid | 0.85 * | 0.67 * | 0.31 NS | 0.72 * |
| Chlorogenic acid | 0.95 * | 0.73 * | 0.44 NS | 0.72 * |
| P-coumaric acid | 0.31 NS | 0.23 NS | 0.46 NS | 0.49 NS |
| Ferulic acid | 0.47 NS | 0.44 NS | 0.62 NS | 0.46 NS |
| Total phenolic acids | 0.89 * | 0.76 * | 0.34 NS | 0.86 * |

* $p \leq 0.05$, NS—statistically insignificant.

Moreover, it was found that among the tested active ingredients, the presence of quercetin was positively correlated with all tested indicators, even with chelating activity (r = 0.79). The chelating activity did not depend on the content of other active components. The presence of most flavonols and acids increased the antiradical activity and reducing properties.

### 4.3. Microbiological Quality

No undesired microflora was found in the tested products since production and after freezing, which proves their safety and guarantees microbiological stability during freezing (Table 5).

**Table 5.** Microbiological characteristic of bread.

| Sample | Total Mesophilic Microorganisms | Total Psychrophilic Microorganisms | Yeasts, Molds | *Listeria monocytogenes* | *Salmonella* | Enterobacteriaceae | Microbial Indicator | | | Coagulase-Positive Staphylococci |
|---|---|---|---|---|---|---|---|---|---|---|
| | | | | | | | Anaerobes | Coli Group | Enterococci | |
| **Microbiological criteria** | $\leq 1.0 \times 10^4$ cfu/g | - | $\leq 1.0 \times 10^2$ cfu/g | 0 cfu/25 g | 0 cfu/25 g | $\leq 1.0 \times 10^1$ cfu/g | - | 0 cfu/g | - | 0 cfu/g |
| PM1 | $1.5 \times 10^1$ cfu/g | $1.0 \times 10^1$ cfu/g | Yeasts $2.0 \times 10^1$ cfu/g - No molds | N/D in 25 g | N/D in 25 g | N/D in 25 g | N/D in 25 g | N/D in 25 g | N/D in 0.1 g | N/D in 0.1 g |
| PM30 | $1.9 \times 10^3$ cfu/g | $1.0 \times 10^1$ cfu/g | Yeasts $1.3 \times 10^1$ cfu/g - No molds | N/D in 25 g | N/D in 25 g | N/D in 25 g | N/D in 0.1 g | N/D in 0.1 g | N/D in 0.1 g | N/D in 0.1 g |
| PK1 | $1.0 \times 10^1$ cfu/g | N/D in 0.1 g | N/D in 0.1 g | N/D in 25 g | N/D in 25 g | N/D in 25 g | N/D in 0.1 g | N/D in 0.1 g | N/D in 0.1 g | N/D in 0.1 g |
| PK30 | $6.9 \times 10^3$ cfu/g | $6.3 \times 10^2$ cfu/g | Yeasts $6.6 \times 10^1$ cfu/g - No molds | N/D in 25 g | N/D in 25 g | N/D in 25 g | N/D 0.1 g | N/D in 0.1 g | N/D in 0.1 g | N/D 0.1 g |

Abbreviation: N/D—not detected.

### 4.4. Sensory Quality

Sweet and sour were dominant among the smell characteristics in both analyzed samples (Figure 3). The insipid smell was perceived to a lower degree, while grassy and foreign smells were perceived to the lowest degree. However, differences in the scent profile of the samples were found. New bread with mulberry was characterized by a higher intensity of sweet and insipid smell, and the control bread by sour smell. In bread with mulberry, the grassy smell was slightly perceptible and in both samples the level of foreign smell was determined at 0.5. Statistical analysis of the results showed that despite differences in the sensation of some of the smell descriptors, the overall lust for bread without and with mulberry was at a similar level.

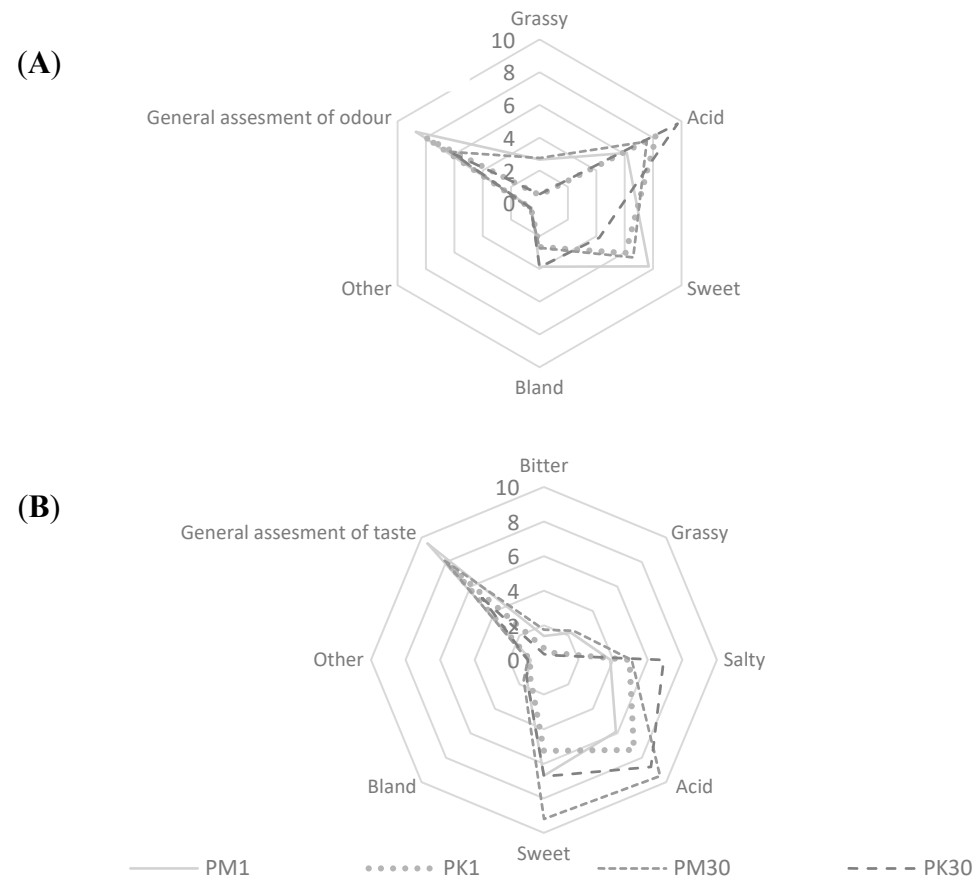

**Figure 3.** Sensory evaluation of bread with the addition of *Morus alba* leaves extract and mulberry alba fruits and leaves extracts ((**A**)- profile of odor, (**B**)- profile of taste).

In terms of the taste of the samples, significant differences were found in the sensation of all the descriptors studied, except for insipid and foreign taste, which were evaluated at a low level, not differing from each other, ranging from 0.69 to 1.35. It was shown that the addition of mulberry to bread significantly increased the feeling of bitter, grassy and sweet taste. On the other hand, the dominant taste in the control bread was sour and salty. As a result of the analysis, a statistically higher evaluation of the overall taste desirability was found for bread with mulberry. It was shown that the sensory profile of the tested samples of mulberry bread changed as a result of the storage. In order to illustrate the evaluation of the extent to which each of the variables is represented by the current set of factors, the results are presented in Figure 4. The further away a given variable is from the center of the circle, the better it is represented by the current coordinate system. The change of smell in the stored samples was not correlated with all relevant taste characteristics. During the storage of the bread, a decrease in the sweet smell and an increase in the grassy smell were observed, while in terms of taste, an increase in the sweet taste and a decrease in the grassy smell were observed. There was

also a correlation between the smell and taste and an increase in the sour taste of the product after the storage.

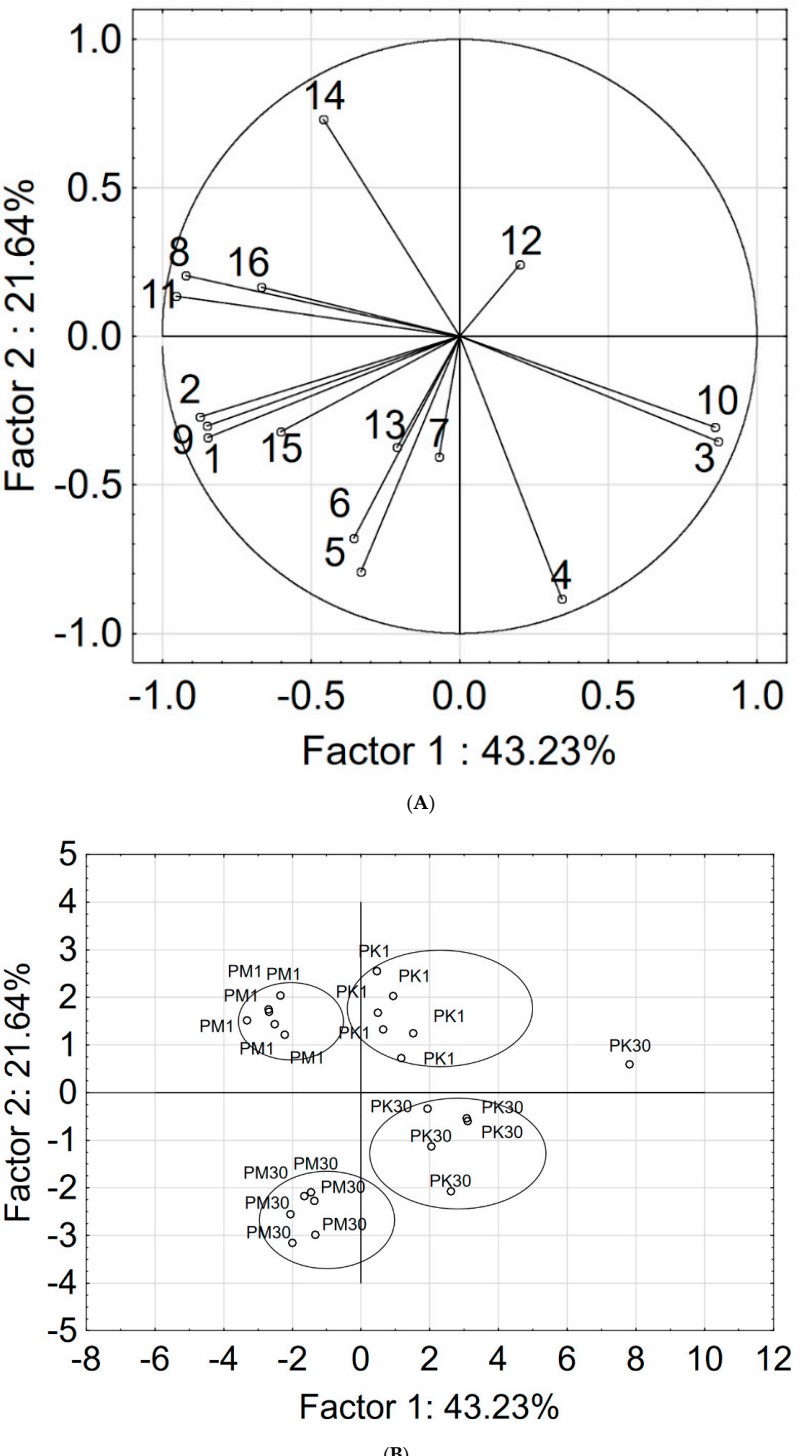

(A)

(B)

**Figure 4.** Principal component analysis (PCA) on the correlation matrix for quality indexes in profile analysis of bread with addition of *Morus alba* fruits and leaves extracts (**A**) for tested samples and principal components projection (**B**). Legend: 1—Bitter (taste); 2—Grassy (taste); 3—Salty (taste); 4—Acid (taste); 5—Sweet (taste); 6—Bland (taste); 7—Other (taste); 8—General assessment of taste; 9—Grassy (odor); 10—Acid (odor); 11—Sweet (odor); 12—Bland (odor); 13—Other (odor); 14—General assessment of odor (odor).

The appearance of sour taste and smell during the storage results, among other things, from the transformation of ferulic acid released during the storage from bound forms, which is decarboxylated and leads to the formation of p-vinylguaiacol. The PCA projection shows significant differences between samples of control bread and mulberry bread, which were stored for 30 days. The differentiation was illustrated in four distant clusters. The stored bread had similar smell and taste characteristics, but the intensity of the characteristics after the storage changed, as can be seen on sample clusters marked as fresh and stored close together.

## 5. Discussion

Technological processes in which both low and high temperatures are used influence the stability of bioactive compounds. Processing processes, among which drying (high temperature) or freezing is the most effective method of preserving the nutritional value and sensory quality of raw materials. In order to maintain active compounds, the process of freezing most plant materials is preceded by blanching. In the case of bread, it is not necessary because the bread before freezing is usually pre-baked and then finally baked. However, raw materials such as fruit, including mulberry fruit, should be blanched to inactivate the enzymes responsible for the oxidation of natural antioxidants [25]. Spray drying and freeze drying (freeze drying) are the most commonly used methods used in the food industry to dry thermolabile substances. Dried mulberry fruit and white mulberry spray-dried extract can be a valuable ingredient in functional foods such as bread. However, it is true that during spray drying the temperature reaches 185 degrees, and according to different studies the polyphenol content of the dried extract could possibly be about 20% higher if a carrier such as maltodextrin was used [26]. The literature on the subject has repeatedly emphasized the beneficial properties of mulberry and the possibilities of its use in food technology. However, there are no reports to date that have determined the influence of mulberry addition on the quality and properties of frozen bread. Earlier studies indicated the possibility of using mulberry fruit as an ingredient in muesli [4]. It was observed that in the samples with extract addition, the polyphenol content was stable in comparison with the control sample, which showed a statistically significant decrease in total polyphenol content after baking. Therefore, it can be assumed that the addition of mulberry not only increased the content of individual phenolic acids and flavonoles in new bread, but also that these compounds stabilized the presence of other polyphenolic substances. The stability of antioxidant compounds was confirmed in the results of the cited study, during fat stability tests by Rancimat and Oxidograph method. It was shown that heating of oil with mulberry extract at 110°C does not inactivate its antioxidant activity.

Moreover, the components of mulberry leaf extract showed antioxidant activity measured by the protective factor. The temperature during bread baking does not exceed 100°C inside the loaf, so most of the mulberry compounds are probably not thermally inactivated. The stability of phenolic acids and flavonols in model systems was also confirmed in other studies, where high stability was demonstrated in a wide temperature range for ferulic acid, the content of which was constant in the temperature range from 25°C to 100°C, and the antioxidant effect of caffeic and sinapic acid was higher at 90°C than 22°C [27,28]. Mulberry fruits have a high antioxidant potential and the ability to chelate metals [29]. The presence of most flavonols and acids increased the antiradical activity and reducing properties. Other authors pointed out that the activity measured by reducing power and ABTS-+ is influenced by the same compounds [30]. The antioxidant effect of mulberry polyphenols was documented many times before, which may have determined the effect in bread samples [6,29]. The bread consisted mainly of wheat flour (38%) and rye flour (26%). The presence of polyphenols in grains was found mainly in the fruit and seed coat, aleurone cells, husk and rarely in the germ. The basic raw material for bread production was whole wheat flour, in which the main polyphenolic compound is tricin (5,7,4'-trihydroxy-3',5'-dimethoxy flavone). This compound and others may have interacted with mulberry polyphenols.

The baking process allows for high purity and, consequently, microbiological stability of the resulting product. However, certain groups of yeasts and molds, acidifying type bacteria as well as

endospore-forming bacteria can survive the temperature of 95 °C , which is achieved during baking in the crumb. The source of contamination may be plant raw materials (cereals) and the environment of the production process (hygienic conditions of the plant), the most common are molds such as Aspergillus, Penicillium and Fusarium and bacteria of the genera Rhizopus, Penicillium and Bacillus [31]. It was found that the breads were characterized by high microbiological purity, which is indicated by the results of analyses performed for the count of indicator microorganisms and the absence of pathogens. Precise determination of the components of sensory quality is a subject of interest for food producers due to the possibility of manufacturing a competitive product with repeatable parameters, meeting growing consumer requirements. The addition of the so-called non-bread flours increases taste and smell qualities of bread, prolongs its freshness and modifies its texture characteristics [27]. The addition of milk and milk products or unusual cereals and cereal products is also frequently used, and literature reports the addition of plant extracts such as grape, rosemary extracts or soya and barley sprouts to bread and cereal products [21,32,33]. All these ingredients must of course have a positive effect on the sensory quality of the food. Data evaluating an effect of rosemary extract on the sensory quality of bread are available in the literature. Costa de Conto et al. indicated that the additive above 2.5 g was negatively perceived by consumers [33]. The bread was characterized by a typical taste identified as rosemary and had a bitter aftertaste. Therefore, other authors suggested that ground leaves of raw materials that are a source of polyphenols could be used instead of extracts. Polyphenols are responsible for the bitterness of the leaves and extracts made from them, tannins form complexes mainly with polysaccharides and proteins, giving bitterness and astringency. Ferulic acid and vanillic acid, which are found in extracts from mulberry leaves and fruit, are responsible for the bitterness of the taste. These acids also give the wrong bitter bean-like taste to protein isolates from oil plants. Fresh mulberry leaves contain flavonol glycosides, mainly 3-O-di- and 3-O-triglucosides of quercetin and kaempferol, as well as C-glycosides of quercetin. These glucosides may affect the bitter but also the sour taste. However, the appropriately selected composition of the raw materials of the bread being tested, despite the raw materials which may have a negative influence on sensory qualities, allowed to obtain an attractive functional bread with mulberry.

## 6. Conclusions

The addition of extracts from mulberry leaves and fruits to fresh bread and bread stored in freezing conditions did not adversely affect the sensory characteristics of samples with high polyphenol content and antioxidant activity. Finally, it should be concluded that the bread is a product which can be an attractive matrix as a carrier of mulberry bioactive substances. The use of mulberry leaf extract and mulberry fruit extract in the production of bread does not alter the microbiological quality during the freezing storage of the final product, which may be an interesting alternative in the diet.

**Author Contributions:** Conceptualization, formal analysis and methodology: J.K.-C.; formal analysis: D.S., O.S., S.B., A.T.; writing—original draft preparation: M.D.; supervision: P.S. All authors have read and agreed to the published version of the manuscript.

**Funding:** The publication was co-financed within the framework of the Ministry of Science and Higher Education program as "Regional Initiative Excellence" in years 2019–2022, project number 005/RID/2018/19 and by statutory funds of the Department of Gastronomy Sciences and Functional Foods of Poznan University of Life Sciences, grant number 506.751.03.00.

**Conflicts of Interest:** The authors declare no conflict of interest.

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
