# Peer review of "The Effects of Morus alba L. Fortification on the Quality, Functional Properties and Sensory Attributes of Bread Stored under Refrigerated Conditions"

_sustainability, doi:10.3390/su12166691_

Round 1
Reviewer 1 Report
English language:
Line 28: morcetin?
Line 30: caffeic acid
Line 141: contractual points?
Line 169: flavonols instead of flavonoles
Lines 264-266: the phrase lacks a predicate
General comments:
The present study aims at evaluating the effects of Morus alba on antioxidant and sensory attributes of bread stored under freezing conditions.
I found the paper to be fairly written overall. The objectives of the study are clearly stated and identified in the Introduction section. On the other hand, the methods are not carefully described. Also, I am unconvinced about the way the authors set the experimental design of the study. In addition, the presentation of the results is somewhat confusing and misleading. This raises a lot of doubt about the inferences and conclusions drawn by the authors.
These observations are detailed below.
Therefore, I recommend undergoing major revision.
Specific comments:
Line 72, Section 3.2: From the manner this section is written, the readers would infer that only one concentration of each M. alba leaves extract and fruit was used in the experiment. And this is corelated with Table 1, which confirms it. I would have expected the authors assayed more than one concentration, which would have rendered the statistical analysis and the associated inferences more meaningful.
This issue is closely linked with my next comment.
Line 109, Section 3.5: I suggest the authors provide a brief description of the method. As presented in the cited study, Kobus-Cisowska et al. (2020), the chelating activity is given as the mean of three replicates (n=3) and standard deviation.
In this context, the presentation of the results corresponding to this analysis is very unclear – Figure 1. Panel a) lacks the legend. Also, the legend for panel b) does not match the used symbols. In fact, the presentation of the results doesn’t seem to be appropriate.
From the phrase at Lines 181-182 citing Figure 1, the reader would infer that the authors evaluated more than one concentration.
I suggest revisiting these sections of the manuscript in order to properly provide an overview of the obtained results.
Line 154, Section 3.8: It should be specified how the PCA analysis was conducted in terms of the employed data matrix, either the correlation matrix or the covariance matrix.
For example, if you have variables with widely varying scales for raw, untransformed data, the use of correlation as an input to PCA would standardize each of the variables (to mean 0 and standard deviation 1). However, if all of your data are from the same platform with similar range and scale, which presumably is the case in the present study, then using correlation will throw out a tremendous amount of information.
Therefore, it would be useful for the readers to be given this information.
The PCA analysis also needs to be presented in more detail, and accompanied by the used factor rotation method, if any (as can be inferred from Figure 4). Moreover, panel A of Figure 4 was most probably constructed based on the sensory responses from the 20 trained evaluators. Is this correct? If yes, please provide this information.
Table 2: lacks the title.
Figure 2: does the use of “s.m” in the figure legend DPPH (µM Tx/g s.m) mean dry matter?
Table 5: I presume jtk/g stands for cfu/g. Please modify accordingly.
Line 267: “stored for five days” – this should have been indicated previously in the Methods section.
Lines 294-295: similar to the observation above, the additives inclusion levels should have been indicated prior to the reader suddenly discovering them all the way here.
Author Response
Dear Reviewer,
Thank you for your thorough review. All comments and recommendations have been included in the revised version of the manuscript. Changes to the manuscript were made in the content of the publication. The comments listed in the review are referenced below.
We hope that the current state of the manuscript meets the standards of the Journal and that publication will be possible.
Sincerely,
Marcin Dziedzinski
---------------------------------------
Comments and responses:
English language:
Line 28: morcetin?
Line 30: caffeic acid
Line 141: contractual points?
Line 169: flavonols instead of flavonoles
Lines 264-266: the phrase lacks a predicate
Response: We corrected these issues.
Specific comments:
Line 72, Section 3.2: From the manner this section is written, the readers would infer that only one concentration of each M. alba leaves extract and fruit was used in the experiment. And this is corelated with Table 1, which confirms it. I would have expected the authors assayed more than one concentration, which would have rendered the statistical analysis and the associated inferences more meaningful.
Response: Unfortunately we used only single concentrations in this experiment.
This issue is closely linked with my next comment.
Line 109, Section 3.5: I suggest the authors provide a brief description of the method. As presented in the cited study, Kobus-Cisowska et al. (2020), the chelating activity is given as the mean of three replicates (n=3) and standard deviation.
Response: We corrected that.
In this context, the presentation of the results corresponding to this analysis is very unclear – Figure 1. Panel a) lacks the legend. Also, the legend for panel b) does not match the used symbols. In fact, the presentation of the results doesn’t seem to be appropriate.
Response: We corrected the figure.
From the phrase at Lines 181-182 citing Figure 1, the reader would infer that the authors evaluated more than one concentration.
Response: We corrected that issue.
I suggest revisiting these sections of the manuscript in order to properly provide an overview of the obtained results.
Line 154, Section 3.8: It should be specified how the PCA analysis was conducted in terms of the employed data matrix, either the correlation matrix or the covariance matrix.
For example, if you have variables with widely varying scales for raw, untransformed data, the use of correlation as an input to PCA would standardize each of the variables (to mean 0 and standard deviation 1). However, if all of your data are from the same platform with similar range and scale, which presumably is the case in the present study, then using correlation will throw out a tremendous amount of information.
Therefore, it would be useful for the readers to be given this information.
The PCA analysis also needs to be presented in more detail, and accompanied by the used factor rotation method, if any (as can be inferred from Figure 4). Moreover, panel A of Figure 4 was most probably constructed based on the sensory responses from the 20 trained evaluators. Is this correct? If yes, please provide this information.
Response: We used correlation matrix, we added needed informations.
Table 2: lacks the title.
Response: We corrected that.
Figure 2: does the use of “s.m” in the figure legend DPPH (µM Tx/g s.m) mean dry matter?
Response: It should be dm for dry matter, We corrected that.
Table 5: I presume jtk/g stands for cfu/g. Please modify accordingly.
Response: We corrected that.
Line 267: “stored for five days” – this should have been indicated previously in the Methods section.
Response: That was a mistake, it should be thirty days as indicated in methods. We corrected that.
Lines 294-295: similar to the observation above, the additives inclusion levels should have been indicated prior to the reader suddenly discovering them all the way here.
Response: This is probably issue made by translation, we refered to the cited study in these lines, not to experiments in the manuscript. We changed that.
Reviewer 2 Report
The Manuscript entitled: "The effects of Morus alba fortification on the antioxidant activity, polyphenols content and sensory attributes of bread" by Kobus-Cisowska et al., describe an interesting study dealing with food fortification. However, before further consideration, the following concerns must be clarified.
1) Title and Abstract (or at least one of these two) must be modified. In fact, according to the Abstract section, the aim of the work is evaluating the impact of freezing storage of bread following the addition of mulberry extracts; however, this information is not clearly presented in the Title. Please, revise.
2) Please, add more references in the Introduction section about food (in this case bread) fortification with extracts rich in bioactive compounds.
3) Any information regarding a potential negative effect of the different heat treatments (such as spray drying) on some thermolabile compounds characterizing the concentrated white Mulberry extracts? Polyphenols are very sensitive to heat conditions.
4) Why authors did not used an HPLC coupled with Mass Spectrometry approach? In this case, the polyphenol characterization is really poor and limited to the standard compounds used. Can the authors provide an untargeted phenolic profiling of their extracts?
5) The authors used several in vitro spectrophotometric assays to assess antioxidant potential. However, these assays are not representative of real conditions and particularly sensitive to other reducing agents (such as proteins). Have the authors purified the extracts in order to remove large biomolecules and proteins?
6) Why authors used 30 days storage as sufficient enough time? Any reference to justify this choice?
Author Response
Dear Reviewer,
Thank you for your thorough review. All comments and recommendations have been included in the revised version of the manuscript. Changes to the manuscript were made in the content of the publication. The comments listed in the review are referenced below.
We hope that the current state of the manuscript meets the standards of the Journal and that publication will be possible.
Sincerely,
Marcin Dziedzinski
---------------------------------------
Comments and responses:
1) Title and Abstract (or at least one of these two) must be modified. In fact, according to the Abstract section, the aim of the work is evaluating the impact of freezing storage of bread following the addition of mulberry extracts; however, this information is not clearly presented in the Title. Please, revise.
Response: We corrected that.
2) Please, add more references in the Introduction section about food (in this case bread) fortification with extracts rich in bioactive compounds.
Response: We added more references.
3) Any information regarding a potential negative effect of the different heat treatments (such as spray drying) on some thermolabile compounds characterizing the concentrated white Mulberry extracts? Polyphenols are very sensitive to heat conditions.
Response: We added several new references to the discussion.
4) Why authors did not used an HPLC coupled with Mass Spectrometry approach? In this case, the polyphenol characterization is really poor and limited to the standard compounds used. Can the authors provide an untargeted phenolic profiling of their extracts?
Response: The research was a part of a large experimental model where the aim was to demonstrate the possibility of using mulberry as an ingredient to enrich bread. The authors only had HPLC and no MS
5) The authors used several in vitro spectrophotometric assays to assess antioxidant potential. However, these assays are not representative of real conditions and particularly sensitive to other reducing agents (such as proteins). Have the authors purified the extracts in order to remove large biomolecules and proteins?
Response: The paper demonstrates the application possibilities and benefits of using commercially available raw materials such as dried mulberry fruit or white mulberry leaf extract for industrial use to enrich bread. The work is of an application nature. Thus, it was not purified from proteins
6) Why authors used 30 days storage as sufficient enough time? Any reference to justify this choice?
An own interview was made in the scope of determining the storage time for baking and the storage time of frozen bread in households. The research model with a 30-day shelf life was estimated
Reviewer 3 Report
The manuscript is interesting, easy to follow and well structured. My only remark to the Authors is to add additional information, or possibly rearrange the manuscript to put focus on the sustainability issue. This could be done to stress up more the environmental impact of the Morus parts, or green fingerprint in production of bread. This is up to the Authors.
For this reason, I will recommend a major revision, but Authors could consider it as a minor since they have to move a focus a bit closer to sustainability.
Author Response
Dear Reviewer,
Thank you for your thorough review. All comments and recommendations have been included in the revised version of the manuscript. Changes to the manuscript were made in the content of the publication.
We supplemented the manuscript with the required information about sustainability in the introduction, summary and discussion section.
We hope that the current state of the manuscript meets the standards of the Journal and that publication will be possible.
Sincerely,
Marcin Dziedzinski
Round 2
Reviewer 1 Report
I believe that the revised manuscript has been improved significantly and now warrants publication.
Reviewer 2 Report
The Manuscript has been substantially improved and can be accepted for publication.
Reviewer 3 Report
The manuscript was significantly improved.
I have no further comments.